# OpenReview forum: "Exploring Memorization and Copyright Violation in Frontier LLMs: A Study of the New York Times v. OpenAI 2023 Lawsuit"
_NeurIPS.cc/2024/Workshop/SafeGenAi — SafeGenAi Poster_

### Official Review · Reviewer_cF8E · 2024-10-08
**In depth analysis of the memorization of LLMs, specifically focused on New York Times articles. The paper is not only relevant but the analysis is also thorough, well described and correctly discussed.**

**Rating:** 7
**Confidence:** 4

**Review:**

Summary:
An extensive study on the current New York Times lawsuit regarding the memorization of LLMs is performed. 4 different models from different providers are compared on 3 attacks using 5 metrics. These analysis reveals that while LLMs tend to memorize articles, the claims made by the New York Times exaggerate the extent to which this happens, furthermore OpenAIs model memorize less than rival models.

Strengths:
 - Extensive analysis of a particularly relevant subject (considering the lawsuit)
 - Exhaustive experiments justifying the validity of the claims
 - Very well written text, making the paper not only easily readable but also clear and understandable

Weaknesses:
 - No objections

Review summary:
In depth analysis of the memorization of LLMs, specifically focused on New York Times articles. The paper is not only relevant but the analysis is also thorough, well described and correctly discussed.

---

### Official Review · Reviewer_6mUM · 2024-10-09
**Review of "Exploring Memorization and Copyright Violation in Frontier LLMs: A Study of the New York Times v. OpenAI 2023 Lawsuit"**

**Rating:** 6
**Confidence:** 4

**Review:**

The authors explore the memorization capabilities of several LLMs in response to copyright concerns within training data. This has important implications as it can help identify vulnerabilities in LLMs regarding copyrighted texts.

## Pros
- An in-depth introduction is given to the legal background behind the NY Times's lawsuits.
- Many factors are explored regarding memorization performance, including the impact of using certain metrics.

## Feedback
- While the legal background is helpful to the reader, some parts—such as "legal aspects of memorization"—might be better in the appendix.
- In line 177, the authors state they could not reproduce the NY Times's claims and propose two reasons. However, they ignore those reasons and state another reason—that OpenAI is indeed filtering its outputs—lines 301 - 302. If OpenAI filters its outputs more aggressively due to the lawsuit, it will affect the comparison between the LLMs' memorization capabilities. It is not specified if the other LLMs have this feature. This could imply the authors are evaluating OpenAI's output filter (influenced by the lawsuit), rather than ChatGPT itself. A more appropriate comparison would be between LLM providers that do not filter outputs or are trained to not regurgitate texts -- i.e. LLMs without an external filter.

---

### Official Review · Reviewer_37X7 · 2024-10-10

**Rating:** 5
**Confidence:** 2

**Review:**

- This is a good paper; however, results such as the impact of model size on memorization are based on previous works. ("While finding that the Lawsuit Articles are far more memorized than the baseline sample of news articles, we reproduce known results that model size increases memorization ability.")
- Additionally, review the findings of the paper titled "Spotting LLMs with Binoculars: Zero-Shot Detection of Machine-Generated Text" for insights on mitigating memorization of sensitive data, as well as broader applications.
- From my understanding of the paper, there were no exact matches. Does this pose any legal risks for LLM providers like ChatGPT?